# What are patients' preferences for revision surgery after periprosthetic joint infection? A discrete choice experiment

Fran E Carroll,[1] Rachael Gooberman-Hill,[2] Simon Strange,[2] Ashley W Blom,[2] Andrew J Moore [2]

¹Population Health Sciences, Bristol Medical School, University of Bristol, Bristol, UK
²Translational Health Sciences, Bristol Medical School, University of Bristol, Bristol, UK

**Correspondence to**
Dr Andrew J Moore;
a.j.moore@bristol.ac.uk

## ABSTRACT

**Objectives** Understanding patients' preferences for treatment is crucial to provision of good care and shared decisions, especially when more than one treatment option exists for a given condition. One such condition is infection of the area around the prosthesis after hip replacement, which affects between 0.4% and 3% of patients. There is more than one treatment option for this major complication, and our study aimed to assess the value that patients place on aspects of revision surgery for periprosthetic hip infection.

**Design** We identified four attributes of revision surgery for periprosthetic hip infection. Using a discrete choice experiment (DCE), we measured the value placed on each attribute by 57 people who had undergone either one-stage or two-stage revision surgery for infection.

**Setting** The DCE was conducted with participants from nine National Health Service hospitals in the UK.

**Participants** Adults who had undergone revision surgery for periprosthetic hip infection (N=57).

**Results** Overall, the strongest preference was for a surgical option that resulted in no restrictions on engaging in valued activities after a new hip is fitted (β=0.7). Less valued but still important attributes included a shorter time taken from the start of treatment to return to normal activities (6 months; β=0.3), few or no side effects from antibiotics (β=0.2), and having only one operation (β=0.2).

**Conclusions** The results highlight that people who have had revision surgery for periprosthetic hip infection most value aspects of care that affect their ability to engage in normal everyday activities. These were the most important characteristics in decisions about revision surgery.

## Strengths and limitations of this study

► This is the first study to quantify the value that patients place on different aspects of revision surgery for prosthetic hip joint infection.
► Using a rigorous process involving empirical qualitative research, we identified four attributes for the discrete choice experiment.
► It was feasible for participants to complete the questionnaires, meaning the results were as expected with patients selecting a combination of options reflecting their preferred choice.
► While the sample size of 57 participants sufficiently powered the analyses presented here, it is too small to conduct any subgroup analyses.
► The majority of the study sample were white, male and educated, so results may not reflect the preferences of the wider surgical population.

## INTRODUCTION

For people with osteoarthritis, hip replacement is a common procedure that aims to improve function and reduce pain. In 2016 over 100 000 hip replacement procedures were conducted in the UK. With an ageing population, rates are predicted to rise.[1] Although successful for many people, approximately 0.4%–1% of patients who have undergone primary hip replacement[2 3] and 2%–3% of patients undergoing revision hip replacement[4] develop deep periprosthetic joint infection (PJI) severe enough to warrant surgical revision. Patients with PJI find it devastating. Symptoms include severe pain, inflammation, discharge from the surgical wound, fever, nausea, malaise, reduction in or loss of function, and dislocation, and if left untreated can lead to disability and/or death.[5]

PJI is extremely challenging to treat. Treatment can include 'debridement, antibiotic treatment and implant retention' (DAIR), used in 7.6% of cases, or more commonly major revision surgery, which involves removal of the infected prosthesis, radical debridement of infected tissue and reimplantation ('fitting') of a new prosthesis with subsequent antibiotic treatment.[4] Revision surgery can be provided in a single operation (one-stage) or as a staged operation (two-stage), where the infected implant is removed and the patient is left without an implant (with or without a temporary spacer) while receiving antibiotic

treatment. In the staged process reimplantation of the new implant is delayed, commonly for up to 6 months, but in some instances over 12 months later.[5][6] There is no clear evidence that either a one-stage or two-stage strategy is superior in eradicating infection,[7][8] but qualitative research has shown that two-stage revision places greater burden on patients and families than one-stage revision. This burden is due to the extended period of immobility in between operations, complications associated with the period of immobility and deep psychological distress, with some patients reporting depression and suicidal thoughts.[6]

Surgeons' decisions about which type of revision surgery is most appropriate for an individual patient take into account many factors. Our previous work indicates that decisions are based on a combination of a surgeon's own training and clinical experience of different techniques; the availability of hospital infrastructure such as microbiology services to quickly identify the infecting organism; characteristics of the infecting organism and duration of infection; patient characteristics such as age, comorbidities, frailty and extent of damaged tissue; and published evidence of revision techniques or reports by senior colleagues. Surgeons also considered patients' preferences for surgery, although this could often involve the choice between long-term suppressive antibiotics and surgery.[9] There is no quantitative evidence that characterises patients' preferences for one-stage or two-stage revision surgery in this context, and this is an important area of work to be investigated. The aim of this study is to assess the surgical preferences of patients who underwent revision surgery for prosthetic hip joint infection.

This study is part of a larger programme of research which aims to improve outcomes for patients after PJI. The programme includes a randomised clinical trial comparing one-stage and two-stage revision surgery for prosthetic hip joint infection,[10] within which this study was embedded.

## METHODS
### Discrete choice experiments
We undertook a discrete choice experiment (DCE) to quantify the surgical preferences of patients who underwent revision surgery for prosthetic hip joint infection. The DCE was embedded within a randomised clinical trial that evaluated clinical and cost effectiveness of revision surgery for prosthetic hip joint infection and compared one-stage revision surgery with two-stage revision surgery. Patients who were taking part in the trial were eligible for the DCE.

DCEs are an established method in health services research and have been used to explore a range of health-related services and treatments.[11–14] DCEs involve asking respondents to choose between hypothetical scenarios which describe goods or services, where a 'service' may mean an intervention or an approach to care. The method aims to establish what attributes of that service influence their decision-making and to what extent. This enables quantification of the marginal impact of these attributes. Scenarios within a DCE describe the service of interest (in this case, revision joint replacement) using the same set of attributes, but at different levels in each scenario. Choices between scenarios, or whether to accept or reject a scenario, are used to estimate the influence and value of the different attribute levels.

### Questionnaire development
When a patient faces revision surgery for prosthetic hip infection, a one-stage or a two-stage operation is required, and the decision about which is undertaken is made largely at the discretion of the surgeon, taking into account patient preferences for treatment. The DCE study was designed to engage patients in their preferences for the features associated with these two surgical options, given neither is currently known to be clinically superior in terms of patient outcomes.

Qualitative methods are recommended for DCE attribute development because they enable conceptual development of attributes directly from people's experiences and so better reflect the issues that are likely to matter most to people when making a decision.[15] For this study we developed attributes from our previous qualitative study which explored the impact of prosthetic joint infection and its treatment on patients and their recovery process; one-to-one interviews were conducted with 19 patients with PJI, focusing on the impact of PJI and surgical treatment. The interviews were audio-recorded and transcribed and the qualitative data set was analysed thematically.[6] Levels of attributes were assigned by refining the language used to convey the meaning of the attributes, and particularly where some quantification of an attribute was mentioned by participants during qualitative interview, as illustrated in table 1.[15]

Table 1 describes the attribute and level selection based on the earlier qualitative work, including illustrative quotations and rationale. The final questionnaire comprised a single scenario task in which respondents were asked to imagine this was the first time that they had been offered the two surgical options presented and were asked to consider them carefully before selecting the one (from a pair) they would prefer.

With a 4×4×2×2 design (2 four-level and 2 two-level attributes), a total of 64 different combinations of attribute levels (profiles) are possible. This converts to 32 pairs of profiles, which as a 'full factorial' was considered to be too large for participants to complete.[16] An orthogonal main-effects plan was therefore used to reduce the number of choice sets to 16.[17] Each profile was presented with its pair, and participants were required to select which option they preferred (see figure 1).

### Patient and public involvement
The questionnaire was piloted and refined in collaboration with five patient and public involvement representatives. At an initial meeting of representatives, group

**Table 1** Qualitative support for attributes included in the discrete choice questionnaire

| Attribute | Evidence of attribute inclusion, with pseudonym and surgery type | Levels | Rationale for levels |
|---|---|---|---|
| Number of operations | "There's no way I want two more big operations now at my time of life. You do it all or not at all…I said there was no way I wanted two ops." (Harriet, one-stage)<br>"Of course, emotionally, you want it over and done with as soon as possible…but ultimately that has to be done in the correct way. There's the tortoise and hare situation. There's absolutely no point in rushing ahead if ultimately it's going to fail." (Maggie, two-stage) | 1. One operation.<br>2. Two operations. | Two types of revision surgery are currently provided in healthcare and involve either one or two operations. |
| Ability to engage in valued activities after new hip is fitted | "Fourteen months without a hip joint so it meant that I couldn't drive a car, I couldn't do anything that I'd been used to doing, playing golf or doing anything. Well, I gave up golf actually after the first revision." (Don, two-stage)<br>"But when I for example went to, on holiday recently and I had serious problems getting into the bath to stand in the shower. Because I, I couldn't get in. And in a wet – on a wet surface and that, I'm very conscious of not falling in. I can't afford to fall. So, pain I've got none, stiffness none. Physical function limitations, and that's one of them." (Rory, one-stage)<br>"My aim has always been to get back on my feet as soon as I can, and to walk as good as I can, and that's a big disappointment. I'm not where I think I should have been." (Robert, two-stage) | 1. Can do everything.<br>2. Can do most things.<br>3. Cannot do most things.<br>4. Cannot do anything. | Following revision surgery, the ability to engage in valued activities can be reduced in a major or somewhat more minor way. These levels capture variation in ability identified by patients. |
| Time taken after surgical treatment starts to return to normal activities | "I didn't want to go 14 weeks with effectively one leg. What was worse was not knowing that I had to endure all of those weeks not knowing that I was ever going to get another hip joint back." (Maggie, two-stage)<br>"If I had known how hard it was going to be for her to walk in that interim six months, and if there seemed to have been a reasonable, or a good possibility that the infection would be nuked in a one stage, then that might have been a better outcome for her." (Amelia, two-stage)<br>"I would've thought that if you wanted to go back to work…you wouldn't be very happy [having a 2-stage operation] because you wouldn't be able to do nothing for those six weeks again. Then you'd have to go all through it again after three months of having it done. Six weeks is only a month and a half, and then in another month and a half you're having it all done again." (Jim, one-stage) | 1. 3 months.<br>2. 6 months.<br>3. 12 months.<br>4. 18 months. | These time intervals demonstrate best approximations and a reflection of the need to ensure normal expectations of time taken for soft tissue recovery as expected by surgeons. 18 months is the maximum endpoint that surgeons would suggest for recovery time. |
| Antibiotic side effects | "…the nightmare on heavy antibiotics, toiletry wise. Now I've had to move into a, another bed. My wife and I are married 50 plus years, and I have to have my own room because I'm getting up in the night." (Rory, one-stage)<br>"…I stayed on antibiotics then for, for ever…and after the first [week of antibiotics] I was just dying. I just wanted to lie on the floor and die. I felt so sick. So ill." (Lottie, two-stage)<br>"I felt fine. When I was going to see the surgeon… They'd say, 'How are you today?' I'd say, 'I feel fine, fit! [laughs] I feel well in myself, I eat well.'" (Ray, two-stage) | 1. Affects me a lot.<br>2. Don't affect me much. | Antibiotics are an essential method of attempting to ensure the periprosthetic joint infection is treated and subsequently clear. For some patients, the impact of these antibiotics is significant, while for others there are less severe side effects. |

members were involved in questionnaire development and suggested improvements to its formatting to aid readability, clearer phrasing of the questions to avoid ambiguity and shortening of the instruction leaflet for clarity. At a subsequent meeting, members of the group completed the questionnaire and fed back on their experience. They completed the questionnaire without assistance and felt the instructions were clear, but suggested that key points in the questions should be highlighted and that a contact telephone number should be added to enable participants to seek assistance if needed. For participants in the study, a summary of findings will be sent to those who indicated that they wished to be informed.

| OPTION 1 | | OPTION 2 |
|---|---|---|
| I take antibiotics and the side effects **don't affect me much** *and* I have **two** operations *and* After my new hip is fitted, **I can do most of the things** that I want to do *and* After my surgical treatment starts, it takes **6 months** to get back to the things that I normally do | OR | I take antibiotics and the side effects **affect me a lot** *and* I have **one** operation *and* After my new hip is fitted, **I cannot do most of the things** that I want to do *and* After my surgical treatment starts, it takes **12 months** to get back to the things that I normally do |

Please only tick **ONE** box:

OPTION 1 ☐          OPTION 2 ☐

**Figure 1**  Example profile of the discrete choice experiment.

## Participants

Patients who were recruited into the INFORM (INFection and ORthopaedic Management) randomised controlled trial (ISRCTN10956306) received the discrete choice task after completing the 18-month primary outcome measure. The questionnaire was either posted to participants or completed in person with the assistance of a research nurse during a hospital clinic visit.

## Data analysis

Sample size calculations for DCEs are challenging due to dependence on the true parameter values estimated in the choice model.[18] However, reliable statistical analysis has been demonstrated with sample sizes of 40–120 respondents, and combined with the rarity of PJI (1% of those undergoing hip replacement) a sample size above 50 participants for the DCE was deemed adequate to obtain sufficient data for exploratory analysis and interpretation.[19 20]

Paper questionnaires were distributed to participants as part of the follow-up data collection in the INFORM trial. Questionnaires were returned to the study team between January 2017 and November 2018 and data were entered into a Microsoft Excel spreadsheet. DCE data were effects-coded and analysed using STATA SE V.15.[21 22] The influence of the four attributes on patient choices was analysed using a conditional logit model. As attribute levels are effects-coded, the mean of all coefficients is 0 across each attribute. The effects-coded preference weights (coefficients) produced by the conditional logit model are estimated relative to the mean effect of the attribute, with the p value indicating the statistical significance of 'the difference between the estimated preference weight and the mean effect of the attribute'.[23]

## RESULTS

Of the 80 discrete choice questionnaires provided to trial participants, 57 were returned fully complete (71%) from patients from nine trial sites. Data from six questionnaires were not used in the analysis as they were partially or totally incomplete. Responding participants had a mean age of 70 (range 51–90), 21 (37%) were female, 26 (46%) had undergone one-stage revision, 14 (25%) lived alone and 41 (72%) were retired from work. Table 2 provides further demographic information.

Table 3 shows the regression coefficients and the results from the 57 patients who had fully completed the discrete choice questionnaire.

Analysis indicates that participants had the strongest preference for a surgical option that resulted in the least restrictions on engaging in valued activities after the new hip is fitted, illustrated by the largest preference weight. Other less valued but important preferences were for a surgical strategy that would result in a shorter time after surgical treatment starts to return to normal activities, few or no side effects from antibiotics, and only one operation. The results also suggest that the least restrictions on engaging in valued activities and the shortest time taken to return to normal activity are the individual attributes most valued by patients in this sample. This is indicated by the larger spread of coefficients (ie, more 'value' is placed on changes in these attributes). The most acceptable option was a time period of between 3 and 6 months to return to normal activity; however, there is no clear preference up to 12 months, although 18 months appeared to be significantly disfavoured.

**Table 2** Demographic and clinical characteristics of the respondents of the discrete choice experiment

| Characteristics | Participants (N=57) |
|---|---|
| Age, years, mean (range) | 70 (51–90) |
| Gender, n (%) | |
| Male | 36 (63) |
| Female | 21 (37) |
| Ethnicity, n (%) | |
| White | 55 (96) |
| Black | 1 (2) |
| Mixed | 1 (2) |
| Marital status, n (%) | |
| Married/with partner | 42 (74) |
| Divorced/separated/widowed | 12 (21) |
| Single | 3 (5) |
| Living arrangements, n (%) | |
| With partner/somebody else | 43 (75) |
| Alone | 14 (25) |
| Schooling/education, n (%) | |
| Left at normal school-leaving age | 35 (61) |
| Left after normal school-leaving age | 15 (26) |
| Left before normal school-leaving age | 7 (12) |
| Work situation, n (%) | |
| Retired | 41 (72) |
| Working/sick leave | 14 (25) |
| Unemployed | 2 (4) |
| Surgery received for prosthetic hip joint infection, n (%) | |
| Two-stage revision | 31 (54) |
| One-stage revision | 26 (46) |

## DISCUSSION

This study aimed to investigate and understand patients' preferences for aspects of revision surgery for PJI. Four relevant attributes were identified through earlier qualitative work, and quantitatively patients in this study most value the ability to engage in valued activities and the time taken to return to normal daily activities. This reflects the findings of our previous qualitative work which show that, although both revision strategies impacted greatly on patients and their families, patients receiving two-stage revision surgery experience particularly long periods of immobility and social isolation. This was often followed by a protracted recovery period, which could leave patients much less able than before their primary operation, and some patients experienced profoundly negative psychological effects associated with physical suffering, loss of dignity and independence.[6] It appears that for patients in our sample, 3–6 months to return to normal activity was preferable, although there was no significant difference up to 12 months, but 18 months was disfavoured. This suggests that the acceptable margin of recovery for patients is up to 12 months after their receipt of a new hip joint.

Discrete choice methodology can be challenging for participants because the format of questions is different from standard surveys and items can seem repetitive. We collected feedback from the first 11 participants who completed the questionnaire. We found that those participants who were supported by a research nurse when completing the questionnaire were more likely to complete and return the questionnaire, compared with those who received the questionnaire by post and completed it alone. Participant feedback suggested that the questionnaire was difficult to complete, as the scenarios were similar and appeared to be repetitive. To address this, we amended the questionnaire format and instructions and offered participants support either face to face or by phone with one of the study research nurses. Nurses were then able to

**Table 3** Discrete choice task results from conditional logistic regression

| Attribute | Level | Coefficient | SE | 95% CI | P value |
|---|---|---|---|---|---|
| Ability to engage in valued activities after new hip is fitted | Can do everything* | 0.70 | | | |
| | Can do most things | 0.49 | 0.08 | 0.33 to 0.64 | <0.001 |
| | Cannot do most things | −0.39 | 0.07 | −0.53 to −0.24 | <0.001 |
| | Cannot do anything | −0.80 | 0.13 | −1.05 to −0.55 | <0.001 |
| Antibiotic side effects | Don't affect me much* | 0.22 | | | |
| | Affects me a lot | −0.22 | 0.05 | −0.33 to −0.12 | <0.001 |
| Number of operations | 1* | 0.20 | | | |
| | 2 | −0.20 | 0.07 | −0.35 to −0.06 | <0.001 |
| Time taken after surgical treatment starts to return to normal activities | 3 months* | 0.20 | | | |
| | 6 months | 0.31 | 0.09 | 0.14 to 0.48 | <0.001 |
| | 12 months | −0.06 | 0.05 | −0.15 to 0.04 | 0.22 |
| | 18 months | −0.45 | 0.10 | −0.64 to −0.26 | <0.001 |

*Indicates reference category within attribute.

answer queries about the questionnaire and offer support if needed. The results suggest the group completed the questionnaire in a rational and logical way, meaning the results were as expected with patients selecting an optimal combination of options as their preferred choice. This is an important methodological finding because, although our study demonstrates the feasibility of the DCE method with this population, others conducting similar studies with older, ill populations could consider in advance the need for professional support in the completion of discrete choice questionnaires.

The participants of this study were all individuals who participated in a clinical trial and had already undergone revision surgery for PJI. This meant that the choices that participants were asked to make in the questionnaire were based on scenarios unlikely to reflect their real-life experiences, as in reality such choices would not be available to them since decisions about surgical strategies are based on a wider variety of clinical, surgeon, patient and organisational factors.[9] Also, patients were not being faced with these decisions at the time of questionnaire completion as we decided that it would be unethical to ask patients awaiting treatment to complete a DCE about surgical options in a hypothetical context, at a time when they may be particularly vulnerable. To do so would not meet an ethical standard of protection from harm as it might mean that patients were inadvertently led to believe that there were more or different options available to them than were clinically indicated at that time.

We also found that some participants reported difficulty separating their own recent personal experience of revision surgery from the hypothetical scenarios presented in the questionnaire, as they found it hard to imagine receiving a treatment option that differed from the one that they had received. In terms of methodology, many DCEs are conducted with participants who already have some experience of the treatment attributes under investigation.[12 19 20 24] In our study previous experience meant that participants had some appreciation of the attributes being tested. While the sample size of 57 participants provided sufficient data for the analyses presented here, the sample size was too small to conduct any subgroup analysis to identify whether preferences would differ between participants who had received one-stage or two-stage revision. Similarly, patients who are older, still working or live alone may have had stronger preferences for a one-stage operation than those who are younger, retired and have support at home to cope with a two-stage procedure. The majority of the study sample were also white, male and educated, which means that results may not reflect the preferences of the wider surgical population. Further research could explore preferences in a more diverse population.

This work has provided an initial and important first step in understanding patients' preferences for characteristics associated with revision surgery for periprosthetic infection. It is important that orthopaedic healthcare professionals discuss these attributes with patients when discussing options for surgical and antibiotic treatment for periprosthetic infection.

## Implications

The results of this study offer insight into the preferences of patients for revision surgery and provide valuable information to surgeons from all disciplines. Although factors affecting patient preferences for surgery differ from those valued by medical professionals, consideration should be given to such factors in order to aid shared decision-making where clinical equipoise between options exists.

Previous research using discrete choice approaches has explored patient preferences in a surgical context. This has included examination of preferences for surgical versus non-surgical interventions in a range of conditions, such as oesophageal cancer and ulcerative colitis.[24 25] Some research has explored preferences for conditions in which there are two surgical options, including for ectopic pregnancy, vaginal wall prolapse and osteoarthritis.[26 27] Across all of these conditions, evidence suggests that patients choose options that reduce the need for further surgery or operations, have a shorter recovery time, have lower risk of symptom recurrence and improve ability to preserve existing joint motion in the case of osteoarthritis. Findings from the current study are similar in that patients prefer a surgical option that reduces the number of operations, recovery time and the side effects of antibiotics. However, in the setting of infected joint replacement, patients placed highest value on restoration of function. This was more important to the patients in our study than the number of operations they would have to undergo. Although our study focused on the preferences between one-stage and two-stage surgery, surgeons may need to consider these preferences during shared decision-making about all options for revision surgery for prosthetic hip infection, including the role of debridement with retention of implants (DAIR). Although DAIR is only efficacious in approximately 60% of cases, it is associated with a quicker return to valued activities and improved joint function.[28]

## CONCLUSIONS

Our results show that the most valued characteristics in decisions about revision surgery for prosthetic hip infection were the ability to engage in valued activities and the time taken to return to normal activity. This builds on the findings of our previous qualitative work which shows that, although both revision strategies impact greatly on patients' and their families' everyday lives, patients receiving two-stage revision surgery experience particularly long periods of immobility and social isolation.[6] The desire to return to everyday activities should be taken into account when surgeons are discussing options with patients, particularly when there is equipoise from a surgical perspective about the options available and when the decision is 'preference sensitive'.

**Acknowledgements** The authors would like to thank the participants who gave their time to complete a discrete choice questionnaire. They also would like to thank Dr Erik Lenguerrand for providing demographic data for the study sample and advice on presentation of statistical results, and the study administration and management team, Makita Werrett and Beverley Evanson, for their invaluable support. The authors would also like to thank the INFORM Research Advisory Group (I-RAG) and the Patient Experience Partnership in Research (PEP-R) at the University of Bristol for their contributions to the refinement of the questionnaire.

**Contributors** FEC contributed to the design, analysis and interpretation of data, and drafting the manuscript. RG-H contributed to the conception, design and interpretation of data, and drafting and revising the manuscript. SS contributed to the design, data collection and revising the manuscript. AWB contributed to the conception, design and interpretation of data, and revising the manuscript. AJM contributed to the design and interpretation of data, and drafting and revising the manuscript. All authors gave final approval of the submitted manuscript.

**Funding** This study is funded by the National Institute for Health Research Programme Grants for Applied Research (NIHR PGfAR) programme (grant number: RP-PG-1210-12005) and supported by the NIHR Comprehensive Clinical Research Network (CRN). This study was also supported by the NIHR Biomedical Research Centre at the University Hospitals Bristol NHS Foundation Trust and the University of Bristol. The views expressed are those of the authors and not necessarily those of the NIHR or the Department of Health and Social Care.

**Competing interests** None declared.

**Patient consent for publication** Not required.

**Ethics approval** Ethical approval for the study was granted by the NRES Committee South West - Frenchay on 31 December 2014 (14/SW/1166). All participants provided written informed consent to take part in the DCE study.

**Provenance and peer review** Not commissioned; externally peer reviewed.

**Data availability statement** Data are available upon reasonable request. Anonymised research data will be made available at the University of Bristol Research Data Repository (data.bris) beginning 1 year after publication of the results. Access will be granted to bona fide researchers after the University of Bristol Data Access Committee has approved their request. Please contact data-bris@bristol.ac.uk in the first instance.

**ORCID iD**
Andrew J Moore http://orcid.org/0000-0003-3185-1599

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
