## [Reviewer comments · BMJ Open]

ARTICLE DETAILS

TITLE (PROVISIONAL)	What are patients' preferences for revision surgery after periprosthetic joint infection? A discrete choice experiment
AUTHORS	Carroll, Fran; Gooberman-Hill, Rachael; Strange, Simon; Blom, AW; Moore, Andrew

VERSION 1 – REVIEW

REVIEWER	Catharina G.M. Groothuis-Oudshoorn University of Twente The Netherlands
REVIEW RETURNED	17-Jun-2019

GENERAL COMMENTS	This is a well written article. However I think that wrt the statistical analysis and reporting of the data it can be improved.  - the results in the abstract are vague, i.e. no numbers are given (no effect size) - when you talk about to engage in normal everyday activities what timespan is meant then? This attribute is not completely independent of the other attribute (ability to engage in valued activities after new hip is fitted). - if you see 'fitted' is that directly after the surgery or when the rehabilitation period is successfully finished? - the sample size is very small. There are ways to calculate provisional sample sizes see e.g. the article of M. Jonker, B. Donkers and E. Grob-Bekker (Sample Size Requirements for Discrete-Choice Experiments in Healthcare: a Practical Guide). - no limitations are discussed wrt this small sample size except for the fact that no subgroup analyses can be done. - the sample consists of patients that underwent a surgery, not patients what had an infection isn't it? So therefore the argument on the rarity of the infection wrt the sample size is strange. - Data analysis: 'the coefficients indicate the strength ...: this is not true; all coefficients are relative to each other. Please consult e.g. the article 'Statistical Methods for the Analysis of Discrete Choice Experiments: A Report of the ISPOR Conjoint Analysis Good Research Practices Task Force, Value in Health'. - coefficients are only relative. So I encourage the writers to calculate also e.g. attribute importance measures. Moreover a graph of the results would also be insightfull - p-values on statements given in the results section are missing (e.g. of comparisons within an attribute between levels). - the argument of largest positive coefficient is in principal theoretically not true. - did you not find any DCEs on hip replacements? (I am aware of a few studies).
--

REVIEWER	Aaron Tande Mayo Clinic USA
REVIEW RETURNED	11-Sep-2019

GENERAL COMMENTS	The authors present an important manuscript that tries to further understand patient preferences for medical/surgical treatment of prosthetic joint infection. The methodology used, the discrete choice experiment, is well detailed and easy to understand. The findings, that the strongest preference was for no restrictions on valued activities after completion of surgical procedures, and the relation to other preferences is an important finding and can immediately assist in the Medical/Surgical decision making in the setting of PJI. Overall, I do not have any significant revision requests, other than I think it would be beneficial to have a statistical review from someone with expertise in the methodology used here.
--

VERSION 1 – AUTHOR RESPONSE

Reviewer 1

1. This is a well written article. However I think that with regards to the statistical analysis and reporting of the data it can be improved.

Thank you for your comments about our article, and we are pleased that you found it to be well written. We have made revisions to the text in relation to the data reporting and these are described in turn below.

2. The results in the abstract are vague, i.e. no numbers are given (no effect size)

Thank you, we have amended the abstract to include the coefficients relating to the results.

3. When you talk about to engage in normal everyday activities what timespan is meant then? This attribute is not completely independent of the other attribute (ability to engage in valued activities after new hip is fitted).

Thank you for asking for clarification about the timespan in relation to engagement in everyday activities. Time taken is described in Table 3 (we have now clarified this by adding text indicating months). Timescales are also described on pages 11 (line 11) and 12 (lines 14-17), and Table 1 on page 19. We feel the difference in attributes is sufficient, in the sense that one describes ability and the other time.

4. If you see 'fitted' is that directly after the surgery or when the rehabilitation period is successfully finished?

We think that this comment also refers to the statement “Ability to engage in valued activities after new hip is fitted”. The statement refers to the fitting or reimplantation of the new hip prosthesis. In a one-stage revision a new hip prosthesis is reimplanted straight away. In a two-stage revision process, there is a period of months between the first surgery and the second surgery, when patients do not have a definitive hip implant allowing for treatment with antibiotics and eradication of infection. It is during the second operation that a new hip prosthesis is fitted (reimplanted). The phrasing of the questions were refined with patient representatives and piloted to ensure clarity. Patients completing the questionnaires also had intimate knowledge of the process and so we are confident they understood the phrasing. For the reader’s clarity we have included the following text on page 4 line 1: “reimplantation (“fitting”) of a new prosthesis with subsequent antibiotic treatment”.

5. The sample size is very small. There are ways to calculate provisional sample sizes see e.g. the article of M. Jonker, B. Donkers and E. Grob-Bekker (Sample Size Requirements for Discrete-Choice Experiments in Healthcare: a Practical Guide).

Thank you, we agree the sample size is small and the limitations of this are discussed on page 14 (lines 1-8). Whilst no formal minimum sample size calculation was conducted, this DCE study was

essentially exploratory in nature and a 'first step' to using this methodology within the challenges of both the population and rarity of the subject area and we do feel sufficient numbers were achieved to provide meaningful results. We were also limited by the number of participants in the main trial who were willing to participate in the DCE, but still achieved a 71% response rate. We highlight the challenges of calculating sample sizes for this method on page 8 and have amended the text on line 7 of page 8 to emphasise the exploratory nature of the work.

6. No limitations are discussed with regard to this small sample size except for the fact that no subgroup analyses can be done.

Thank you for this comment. While the sample size provides sufficient data for the analysis presented, it is too small for sub-group analysis. We include some reflection on the sample as there is a lack of diversity because the majority were white, male and educated. Unfortunately, due to the nature of PJI and the fact that it affects only 1% of those having hip replacement, small sample sizes are a limitation of any research on this population. This is explained on page 14 (lines 1-8).

7. The sample consists of patients that underwent a surgery, not patients that had an infection isn't it? So therefore the argument on the rarity of the infection with regard to the sample size is strange. All patients had surgery for prosthetic joint infection. This is stated in the abstract (page 2 lines 9-10, 13 & 18) and throughout the main text e.g. page 5 lines 6-7, page 6 line 13, page 13 line 9-13. We do explain at length (page 13) that all patients had received treatment for infection as part of the trial, and that it is normal for participants to have some experience of the treatment under investigation as shown in other studies (page 13 line 25).

8. Data analysis: 'the coefficients indicate the strength this is not true; all coefficients are relative to each other. Please consult e.g. the article 'Statistical Methods for the Analysis of Discrete Choice Experiments: A Report of the ISPOR Conjoint Analysis Good Research Practices Task Force, Value in Health'.

Thank you for this comment and please accept our apologies for the error. We have amended the manuscript text on: page 8 (lines 14-17) and page 11 (line 5) to clarify and referenced Hauber and colleagues' 2016 paper which you kindly provided above.

9. Coefficients are only relative. So I encourage the writers to calculate also e.g. attribute importance measures. Moreover a graph of the results would also be insightful - p-values on statements given in the results section are missing (e.g. of comparisons within an attribute between levels). The argument of largest positive coefficient is in principal theoretically not true.

Thank you for this comment. We feel our amendments to the manuscript text regarding the 'relative' nature of coefficients and 'largest positive coefficient' in response to your previous comment (number 8) also address this comment. Attribute p values are shown in Table 3 (pages 10 and 11)

Thank you for the suggestion of possibly including further calculations and presentation of the results, but following the second (infectious disease clinician) reviewer's comments that 'The methodology used, the discrete choice experiment, is well detailed and easy to understand' and 'The findings, that the strongest preference was for no restrictions on valued activities after completion of surgical procedures, and the relation to other preferences is an important finding and can immediately assist in the Medical/Surgical decision making in the setting of PJI' we feel that the results presented are easy to interpret and meaningful to a range of audiences. The results offer insight into patient preferences, providing valuable information to surgeons from all disciplines and as such provide a starting point for considerations of these factors to aid shared decision making.

10. Did you not find any DCEs on hip replacements? (I am aware of a few studies).

Thank you, we have added a reference (van Dijk et al, reference number 14), in the introduction where we write about the use of DCE methods in a variety of conditions. The van Dijk article is about risk tolerance for primary hip replacement rather than infection after joint replacement and there are no other studies about PJI of which we are aware.

Reviewer 2

11. The authors present an important manuscript that tries to further understand patient preferences for medical/surgical treatment of prosthetic joint infection. The methodology used, the discrete choice experiment, is well detailed and easy to understand. The findings, that the strongest preference was

for no restrictions on valued activities after completion of surgical procedures, and the relation to other preferences is an important finding and can immediately assist in the Medical/Surgical decision making in the setting of PJI.

Thank you. We are pleased you agree the findings are important.

12. Overall, I do not have any significant revision requests, other than I think it would be beneficial to have a statistical review from someone with expertise in the methodology used here.

Thank you, we hope that our responses to the comments from Reviewer 1 adequately address questions that a methodological expert has about the study and its presentation.

We do hope that we have satisfactorily answered all of the comments and look forward to hearing from you.